# Interactivity and Trust as Antecedents of E-Training Use Intention in Nigeria: A Structural Equation Modelling Approach

**DOI:** 10.3390/bs7030047

**Published:** 2017-07-18

**Authors:** A. U. Alkali, Nur Naha Abu Mansor

**Affiliations:** 1Faculty of Management, Universiti Teknologi Malaysia, Johor 81310, Malaysia; nurnaha@utm.my; 2School of Management and Information Technology, Modibbo Adama University of Technology, Yola 640221, Nigeria

**Keywords:** e-training, interactivity, trust, perceived ease of use, perceived usefulness

## Abstract

Background: The last few decades saw an intense development in information technology (IT) and it has affected the ways organisations achieve their goals. Training, in every organisation is an ongoing process that aims to update employees’ knowledge and skills towards goals attainment. Through adequate deployment of IT, organisations can effectively meet their training needs. However, for successful IT integration in training, the employees who will use the system should be positively disposed towards it. This study predicts employees’ intention to use the e-training system by extending the technology acceptance model (TAM) using interactivity and trust. Methods: Two hundred and fourteen employees participated in the study and structural equation modelling was used in the analysis. Results: The findings of the structural equation modelling reveal that interactivity, trust, perceived usefulness and perceived ease of use have direct and positive effects on employees’ intention to use e-training. It was also shown that perceived ease of use had no effects on perceived usefulness, while trust has the strongest indirect effects on employees’ intention. In addition, the results of Importance-Performance Map Analysis (IPMA), which compares the contributions of each construct to the importance and performance of the model, indicate that to predict intention to use e-training, priorities should be accorded to trust and perceived usefulness.

## 1. Introduction

Providing a competent workforce is the desire of every modern organisation as its success largely hinges on the performance of its human resource [1]. To have competitive advantage and comply with the demands of the emerging global labour market infrastructure, organisations must create a mechanism that ensures the availability of a workforce with the prerequisite knowledge, skills, and ability to effectively deliver within the existing constraints of global competition. This can be achieved through the provision of extensive training that provides employees with updated knowledge of modern changes in technologies and socio-economic set up in the world of competition [2]. Sustaining the traditional method of training is becoming less attractive to organisations due its costly nature [3,4] and the existence of alternative and more sustainable options provided by technology [5,6]. According to Kamal, Aghbari [7], the demand for alternative methods for learning is increasing exponentially, and that the demand for corporate electronic learning is on the increase, as many corporations have adopted e-learning for employee training and learning. However, organisations should be cautious in their drive towards technology integration in training. To ensure success of an e-training system, the employees, who are major stakeholders, should be able and willing to use such systems, even though options available to them in a mandatory use environment are very limited. The importance of assessing intention to use information technology (IT) prior to its implementation has been recommended [8]. Although previous studies on intention to use technology exist in the extant literature, most of these studies have investigated intentions in the fields of e-commerce, e-banking, mobile banking, consumer behaviour, websites and cloud services [9,10]. The roles played by interactivity and trust as determinants of intention have been established in the aforementioned areas [9,11,12]. However, research in the field of e-training is generally scarce in the extant literature. Specifically, no study has empirically investigated the factors influencing intention to use e-training in public organisations [13]. In addition, no study has investigated the roles of interactivity and trust in e-training intentions. This paper presents empirical findings on the influence of interactivity and trust on intention to use e-training using TAM, in the context of Nigerian public universities, in order to fill the gaps in the literature.

### 1.1. Theoretical Foundation

This study uses the technology acceptance model (TAM) [14] as its underpinning theory. According to the TAM, user perceptions of usefulness (PU) and ease of use (PEOU) determine the attitudes toward using the system, and attitudes toward using the system determine behavioural intentions, which in turn, lead to actual system use. Perceived usefulness is the degree to which a person believes that using a particular system would enhance his/her job performance, while perceived ease of use is the degree to which a person believes that using a particular system would be free of physical and mental effort [14]. According to Davis (1989) [14], the sequential relationship of belief–attitude–intention–behaviour in TAM, enables us to predict the use of new technologies by users (see Figure 1).

Since its inception in 1989, the technology acceptance model has been widely used by researchers in investigating behavioural intentions and acceptance of technology across different fields of study and contexts. The reason advanced for its acceptance among researchers is the degree of its flexibility, which can be modified, based on the purposes of the study, enabling it to be extended [15]. Many studies have used the TAM to investigate intention to use technology and found it to be effective in predicting intention [9,10,12,16,17,18]. Therefore, this study aims at extending the TAM, by examining the influence of interactivity and trust on the intention to use e-training.

### 1.2. Conceptual Framework and Hypotheses

Based on the TAM, perceived usefulness and perceived ease of use are used to explain intention to use e-training. In the context of this study, perceived usefulness and perceived ease of use refer to employees’ perceptions of e-training in the universities. Interactivity, is a formative second-order construct, which means that it is formed by other constructs or dimensions. In this study, interactivity is formed by three other dimensions namely, active control, two-way communication, and synchronicity, based on the modified Liu’s interactivity dimensions [19]. The trust construct is adapted from Gefen, Karahanna [11]. Both constructs denote the external variables influencing intention to use e-training through perceived usefulness and perceived ease of use.

#### 1.2.1. Perceived Ease of Use

Perceived ease of use is one of the main constructs of the TAM. In this study, perceived ease of use refers to the degree to which an employee believes that using e-training system will be easy to operate, understandable, interactive, and flexible. Previous studies have reported factors such as self-efficacy, experience, understandability, interactiveness, flexibility and facilitating conditions, to have influenced perceived ease of use [20,21,22]. Thus, the ability of employee to use the e-training system will be affected by his/her level of self-efficacy, ease of understanding, and flexibility of the system. On the other hand, employees PEOU will improve his/her perceptions of the usefulness of the e-training system in terms of task performance, skills, and reward emphasised. Equally, PEOU use is likely to remove the fear that usually comes with the introduction of a new system and reinforce employees’ trust towards e-training. Prior studies have reported that most research conducted in the field of IT have confirmed the existence of significant and positive relationships between perceived ease of use, attitude and intention to use e-learning [23]. Likewise, PEOU was reported to have had significant and direct effects on PU [24,25,26]. In addition, the indirect influence of PEOU on intention through PU has also been established [24,27,28,29]. On the contrary, PEOU was found to have no effects on PU [30] and on intention [31]. Considering the evidences above, the researchers propose the following hypotheses:

**Hypothesis** **1.**Perceived ease of use positively and significantly affects employees’ intention to use e-training in Nigerian public universities.

**Hypothesis** **2.**Perceived ease of use positively and significantly affects employees’ perceived usefulness of e-training in Nigerian public universities.

**Hypothesis** **3.**Perceived ease of use influences the effects of interactivity on intention to use e-training in Nigerian public universities.

**Hypothesis** **4.**Perceived ease of use influences the effects of trust on intention to use e-training in Nigerian public universities.

#### 1.2.2. Perceived Usefulness

Perceived usefulness is another important construct predicting intention in the TAM. According to Li and Huang [32], PU in the TAM is the main belief factor determining behavioural intention to use an information technology. It refers to the degree to which an employee believes that using e-training would enhance his/her skills, task accomplishment, productivity, and make work easy and useful. This implies that the more an employee views e-training system as capable of positively affecting his/her personal skills and work performance, then the employee is likely to view a training as very useful worth using. Previous studies have reported PU as an important determinant of attitude and behavioural intention [33]. The influence of the construct on intention to use technology was also established [34]. Similarly, PU was reported to have had a direct and positive effect on employees’ intention to use web-based training [10]. Likewise, PU was found to have predicted behavioural intention to use e-portfolio [20,35] and a strong determinant of adoption [36]. Therefore, building on the above empirical evidence, and in accordance with TAM, the study postulates that PU plays a significant role of influencing employees’ intention to use e-training systems in Nigerian public universities. Hence, the following hypotheses are proposed.

**Hypothesis** **5.**Perceived usefulness positively and significantly affects employees’ intention to use e-training in Nigerian public universities.

**Hypothesis** **6.**Perceived usefulness influences the effects of interactivity on intention to use e-training in Nigerian public universities.

**Hypothesis** **7.**Perceived usefulness influences the effects of trust on intention to use e-training in Nigerian public universities.

**Hypothesis** **8.**Perceived usefulness influences the effects of perceived ease of use on intention to use e-training in Nigerian public universities.

#### 1.2.3. Interactivity

One of the established weaknesses of e-training is the absence of face–to-face interaction [37]. Emphasising on the importance of interaction, Moore and Kearsley [38] have opined that interaction is relevant to all forms of learning, whether such learning involves technology or not. There are many types of interaction in a technology based learning environment in the extant literature. These include, among others: learner–tool interaction [39], learner–task interaction [40], the learner self, learner–learner, learner–instructor, learner–content, and learner–interface types [41]. In this study, interactivity refers to the extent to which two or more parties communicating in an e-training environment can act on each other, on the communication medium, and on the messages, and the degree to which such influences are synchronised. Interactivity in this study is a formative construct with indicators that are fundamental parts of it namely, active control, two-way communication, and synchronicity. Active control refers to the employee’s ability to voluntarily participate in and instrumentally influence a communication; two-way communication refers to the mutual communication: (1) between users and users or (2) between users and messages on the internet; and synchronicity refers to the degree to which employees’ contribution to a communication and the responses they receive from the communication are simultaneous [19]. Previous studies have confirmed that interactivity has a positive influence on the attitude of users and their use intentions [42] and increases intention to use e-learning [43]. Similarly, past studies have reported that interactivity as an exogenous variable has positively influenced perceived usefulness and ease of use [44,45]. Furthermore, Macy and Skvoretz [46] have demonstrated in a simulation study that high levels of interaction leads to trust. Based on the evidence above, the researchers opine that interactivity is important to employees’ use of e-training system in universities. Accordingly, the following hypotheses are formulated:

**Hypothesis** **9.**Interactivity positively and significantly affects employees’ intention to use e-training in Nigerian public universities.

**Hypothesis** **10.**Interactivity positively and significantly affects employees perceived usefulness in Nigerian public universities.

**Hypothesis** **11.**Interactivity positively and significantly affects employees perceived ease of use.

#### 1.2.4. Trust

To reduce uncertainty and get anticipated commitment from employees, trust is argued to be the key [47]. It has been argued that organisations first build trust in their employees before they become affectively committed [48]. For instance, in using an e-training system, issues relating to security of personal information, feedback on e-training performance, safety of e-training platform, terms and conditions for using the university’s e-training system, integrity of people to interact with in the course of e-training process, absence of face to face interaction, and fear of not achieving much during e-training etc., could raise some concerns among employees and might influence their trust in the e-training system. In this study, trust is defined as the extent to which employees consider the e-training system and the training participants as secure, trustworthy, considerate, and in their best interest. Building trust may help in improving perceived usefulness and employees’ experience of using a technology. For instance, when employees believe that e-training system has the necessary ability, integrity and benevolence to deliver a positive training and experience to them, they are likely to use it. According to Gefen, Karahanna [11], trust provides the assurance to the users that using the system will result in positive outcomes in the future. In addition, Luhmann [47] opines that trust is key for an organisation to reduce uncertainty and get anticipated the commitment, and further, that trust can reduce risk and uncertainty in trust related behaviours [49]. Yoon [50] has also confirmed the effects of trust on perceived usefulness. The relationship between trust and intention has also been established in the extant literature. For example, trust has been found as an initial prerequisite for users to participate in knowledge transferring and exchanging [51] and plays a significant role in the promotion of knowledge sharing among users [52]. In another study, Yusoff, Ramayah [53] have established that trust had a significant effect on the attitude towards using electronic human resource management (E-HRM). Also, Wang, Ngamsiriudom [54] have established that the relationship between trust and behavioural intention are positively significant. Based on the above assertions and evidence, this study opines that trust will play an important role in influencing employees’ intention to use e-training in Nigerian universities. Accordingly, the following hypotheses are proposed:

**Hypothesis** **12.**Trust positively and significantly affects employees’ intention to use e-training in Nigerian public universities.

**Hypothesis** **13.**Trust positively and significantly affects employees’ perceived usefulness in Nigerian public universities.

**Hypothesis** **14.**Trust positively and significantly affects employees’ perceived ease of use in Nigerian public universities.

## 2. Methods

### 2.1. Respondents

The respondents of this study were drawn from employees of five federal universities of technology in Nigeria. A multi-stage cluster sampling was used and the respondents were selected through systematic random sampling based on strata. After data cleaning and treatment of outliers, a total of 214 usable responses were retained and used for the analysis. Demographically, the sample has more men than women and the majority of the respondents were aged 30 years and above. Likewise, the analysis indicates that the population is an educated one as the majority of the respondents have Masters Degrees and above. In terms of computer/internet experience, the analysis indicates that 13% had less than 1 year experience, 35.2% had 1–3 years, 31.6% had 4–7 years, 13.6% had 8–11 years and 6.6% of the respondents had more 12 years of experience.

In order to conform to ethics requirements, a cover letter was attached to each questionnaire in which the purpose of the study was clearly stated. Also included in the letter were the names, addresses (including email addresses), and institution of the researchers with the hope of increasing the confidence of the respondents and for them to be familiar with whom they were dealing with, as opined by Cooper and Schindler [55]. Information provided by the respondents was treated as confidential, used only for academic purposes, and were not in any way described to allow their identity be revealed. In addition, the researchers used only the combined results in reporting the findings of the study.

### 2.2. Instrument and Data Collection

A 5-Likert scaled questionnaire was designed to measure the elements of the proposed model. The constructs of the model were selected based on the extant literature in order to assess employees’ intention to use e-training. The items for intention construct (INT) were adapted from Venkatesh and Davis [56] and Dix, Ferguson [57]. Items used in measuring trust were adapted from Gefen, Karahanna [11] and from Zhou [58]. The items for perceived usefulness and perceived ease of use were adapted from Davis [14]. Lastly, the interactivity construct, beings a second order formative construct has its items adapted from Liu [19]. The questionnaires were pre-tested by academic experts and professionals with good experience in training and technology management. Copy of the final questionnaire used in the study is provided in Appendix A. A sample of 214 drawn from the five universities of technology that formed the population of the study, and was used in the final data analysis of this study. Stratified random sampling was used and respondents were selected based on systematic random sampling from each strata (university). The questionnaire was finally administered to the employees of the 5 universities and were asked to indicate their agreement or disagreement with the above items as contained in the questionnaire. The data collection was done between January and March 2017.

### 2.3. Data Analysis Technique

SPSS 21.0 and SmartPLS 3.0 were used for statistical analysis of the data collected. The assessment of the Partial Least Square–Structural Equation Modeling (PLS-SEM) and reporting the output, recommendations of Hair Jr, Hult [59] and Ramayah, Cheah [60] were followed.

## 3. Results

### 3.1. Measurement Model

The measurement model was assessed using item loadings, convergent validity, reliability of measure, and discriminate validity. For convergent validity, the researchers first examined the outer loadings of the indicators, which as recommended by Hair Jr, Hult [59] should be 0.708. Secondly, the researchers examined the average variance extracted (AVE) values of all the constructs in the research model and the results as presented in Table 1, show that all the constructs which must meet the recommended minimum requirement of AVE > 0.50 [61]. An item is reliable if its loading was greater than 0.7. The convergent validity was determined using average variance extracted (AVE), which according to Fornell and Larcker [61] must exceed 0.5. Composite reliability and Cronbach’s Alpha were used to assess the reliability of the measures. Normally, the minimum values of composite reliability should be 0.7 [62], and that of Cronbach’s alpha should also be 0.7 [63]. To check discriminate validity, square root of average variance extracted, latent variable correlations, and Heterotrait–Monotrait Ratio of Correlation (HTMT) were used. As rule of thumb, the square root of average variance extracted of each construct should exceed the correlation shared between one construct and other constructs in the model [61]. For cross loadings, the requirement is that the loadings of indicators on the assigned latent variable should be higher than the loadings on all other latent variables [64]. HTMT discriminant validity between two constructs is deemed to be established if the HTMT_0.85_ value is less than 0.85 [65] or HTMT_0.90_ value of 0.90 [66]. The results in Table 2 shows that all the items outer loadings are above the recommended threshold of 0.708 and the AVEs for constructs are above 0.5. In addition, the composite reliability for all constructs have met the threshold of 0.7. Also, the Cronbach’s alpha for all construct were above the recommended minimum of 0.7.

In addition, based on the results in Table 3, the Fornell–Lacker Criterion shows that the square root of average variance extracted of each construct have exceeded the correlation shared between one construct and other constructs. Lastly, the HTMT Criterion also shows that none of the values is greater than 0.90 as recommended by Gold and Arvind Malhotra [66]. Therefore, based on the results in Table 1 and Table 2, the reflective measurement can be said to have met convergent and discriminant validity.

#### Second-Order Formative Construct’s Measurement Model

Having a second-order formative construct requires that its measurement model be assessed separately, as different indices are required to assess the significance and relevance of the formative indicators. To do that, the researchers first assessed the convergent validity, and the redundancy results show a path coefficient of 0.686, which is approximately 0.70 (see Figure 2), which is acceptable to provide support for convergent validity of the formative construct [59].

Likewise, the researchers checked the outer Variance Inflation Factor (VIF) values using PLS algorithm and all the VIF values were below the recommended 3.3 [67]. Lastly, to assess the significance and relevance of the formative indicators, the researcher run basic bootstrapping using 5000 subsamples. The results in Table 4 indicates that the weights of IRAC (*p*-value = 0.01), IRCM (*p*-value = 0.01), and IRSN (*p*-value = 0.013) were significant. Thus, based on the results in Table 3, the formative construct measurement model has achieved validity. Having ascertained the validity and reliability of the reflective and formative construct measurement models, the researchers then tested the structural model.

### 3.2. Structural Model

To test the structural model, collinearity assessment was carried out and the results indicate that both tolerance and VIF are below the threshold of 10 and 5 respectively [68]. This confirms that multicollinearity is not a concern. Partial least squares-structural equation modeling (PLS-SEM) algorithm was then run to obtain path coefficients (the structural model relationships) which represent the hypothesised relationships among the constructs of the study (see Figure 3).

Path coefficients have standardised values between −1 and +1 which values of +1 represent strong positive relationships, while −1 represent strong negative relationships [59]. This value should be significant, and at least at the 0.05 level [69].

#### 3.2.1. Significance of the Relationships among Constructs

The results of the structural model are presented in Table 5. To determine the significance of each of the path coefficient, a basic bootstrapping using 5000 sub-samples was run as recommended by Chin [64]. The results of the bootstrapping also show the significant structural relationships among the research variables and path coefficients.

The results indicate that perceived ease of use (*t* = 2.365, β = 0.132, *p* < 0.018) has positive influence on intention to use e-training which supports the hypothesis that perceived ease of use positively and significantly affects employees’ intention. However, the influence of perceived ease of use on perceived usefulness (*t* = 0.055, β = −0.004, *p* > 0.1) was found to be statistically insignificant to support the hypothesis that perceived ease of use positively and significantly affects employees’ perceived usefulness. Also, the results show that perceived usefulness (*t* = 7.222, β = 0.453, *p* < 0.01) had a strong influence on intention which supports the hypothesis that perceived usefulness positively and significantly affects employees’ intention. Likewise, the results indicate that interactivity has significant influence on intention (*t* = 2.160, β = 0.138, *p* < 0.031) which supports the hypothesis that Interactivity positively and significantly affects employees’ intention. Interactivity also had significant effects on perceived ease of use (*t* = 6.929, β = 0.490, *p* < 0.01) and thus supports the hypothesis that interactivity positively and significantly affects employees perceived ease of use. Its influence on perceived usefulness (*t* = 4.530, β = 0.399, *p* < 0.01) has also been established thereby supporting the hypothesis that interactivity positively and significantly affects employees perceived usefulness. Furthermore, the results demonstrate that trust has significant influence on intention (*t* = 3.467, β = 0.251, *p* < 0.01) which supports the hypothesis that trust positively and significantly affects employees’ intention. The results also indicate that trust has significantly influenced perceived usefulness (*t* = 5.464, β = 0.446, *p* > 0.1) which supports the hypothesis that trust positively and significantly affects employees’ perceived usefulness. Likewise, trust has positively and significantly influenced on perceived ease of use (*t* = 4.178, β = 0.275, *p*< 0.01), thereby supporting the hypothesis that trust positively and significantly affects employees’ perceived ease of use. Therefore, the results have supported all the direct hypothesised relationships between the constructs except that which exist between perceived ease of use and perceived usefulness.

#### 3.2.2. Mediation

To test the mediation effects of perceived usefulness and trust, PLS-SEM bootstrapping was run as recommended by Preacher and Hayes [70]. Mediation is said to be established when results of indirect effect from the confidence interval bias corrected is all positive or all negative. If the results show that zero is not between the lower and upper bound, it means that the indirect effect is not zero [71]. The results of the indirect effects and the confidence interval bias corrected values as shown from Table 6 indicate that perceived ease of use has mediated the influences of interactivity and trust on intention (all *p*-values < 0.01) which support the hypotheses that perceived ease of use influences the effects of interactivity on intention to use e-training and that perceived ease of use influences the effects of trust on intention to use e-training. Likewise, perceived usefulness has mediated the influences of interactivity on intention and that of trust on intention (all *p*-values < 0.01) thus supporting the hypotheses that perceived usefulness influences the effects of interactivity on intention and that perceived usefulness influences the effects of trust on intention. However, perceived usefulness has failed to mediate the influence of perceived ease of use on intention (*p*-value > 0.1). This means that the hypothesis stating that perceived usefulness influences the effects of perceived ease of use on intention to use e-training was not supported.

#### 3.2.3. Importance-Performance Map Analysis (IMPA)

In order to provide better understanding of the most important constructs influencing intention to use e-training, the researcher conducted an Importance-Performance Matrix Analysis using PLS-SEM. According to Hair Jr, Hult [59], the importance is determined by the total effects of the structural model while the performance is determined by the average values of the latent variable. The use of IPMA to identify which construct(s) in the structural model are relatively important and/or have relatively higher performance has been recommended [59]. This analysis is important as it extends findings from PLS-SEM analysis which offers direct, indirect and total relationships and extract the analysis to include another dimension showing the actual performance of each construct. The importance of carrying out IPMA analysis using SmartPLS has been previously recommended [72,73]. The results in Table 7 show the indicators’ importance-performance map which indicate that trust has the highest performance (76.62) and highest total effect (0.588). This is followed by perceived usefulness with 0.505 and 73.17 importance and performance respectively. Perceived ease of use had the least importance (0.152) though had a relatively high performance (73.77).

## 4. Discussion and Implications

### 4.1. Discussion

As mentioned earlier, successful e-training in organisations could be affected by the employee’s ability and willingness to use such system even though options available to the employee in a mandatory use environment are very limited. The present study proposes that interactivity and trust have direct influence on intention to use e-training as well as indirect influence on intention through perceived ease of use and perceived usefulness. Based on this, nine direct hypotheses and five indirect hypotheses were formulated and tested. From the findings, employees are most probably going to use e-training if such usage is easy and causes less fatigue. On the other hand, the insignificant influence of perceived ease of use on perceived usefulness is an important finding of the study as it goes contrary to one of the fundamental relationships of TAM and findings of prior studies that established a direct and significant influence of perceived ease of use on perceived usefulness [74,75]. The reason could be related to the employees’ competence and experience in computer and internet use based on their educational background and experiences in computer and internet use experiences. This most probably be the reason why the employees perceived e-training system to be inherently easy use, thereby rendering the effects of ease of on usefulness as insignificant. This suggests that perceived ease of use is not a predictor of perceived usefulness in an e-training environment. Furthermore, the findings have confirmed the influence of perceived usefulness on intention, suggesting that employees will most probably use e-training in future when they feel it is a useful tool for enhancing their ability, skills, and performance. In addition, the findings of the study have verified that interactivity can have positive influence upon employees’ e-training use intentions which confirms previous findings that demonstrated the influence of interactivity on intention [43,76,77,78,79]. This means that, the extent of interactiveness of e-training system is most likely to create a positive disposition towards e-training among employees in the universities. The employees see the interactiveness of e-training system as providing cushioning effect on absence of face-to-face training as found in the traditional training methods they are used to. Also, the results have confirmed the influence of interactivity on perceived ease of use and perceived usefulness which is consistent with previous findings [80,81]. This implies that interactiveness of the e-training system is very likely to facilitate knowledge sharing among the trainees which can influence how they navigate and make use of the system in the easiest ways. Likewise, the findings show that trust is the strongest predictor of intention to use e-training. This underscores the importance employees attached to trust while intending to use e-training in the future. The positive and significant effects of trust on perceived usefulness and perceived ease of use suggests that the level of trust employees have on e-training could create positive disposition towards it, as the risk of exposing personal information and not achieving much under e-training will no longer be issues to contend with. This also indicates that trust could increase perceptions of employees on the easiness of the e-training system. The findings have also confirmed the mediating powers of perceived usefulness and perceived ease of use in predicting intention to e-training.

### 4.2. Theoretical/Managerial Implications

Looking at the contributions of this study from the theory-testing perspective, the study takes an initial step toward extending and validating the research results from existing studies. The study has empirically investigated the intention to use to use e-training among university employees using the TAM with interactivity and trust factors influencing employees’ intention to use e-training through perceived usefulness and perceived ease of use. This study contributes to the literature by generating empirical evidence that supported the critical role of interactivity and trust as significant determinants of intention to use e-training. The findings reiterated the ability of TAM to explain intention; thus, supporting the extant literature [15,82]. The extension of TAM in this study reveals that, when TAM is applied in the context of e-training use intention, perceived usefulness and perceived ease of use remain strong determinants of use intention as previously established in the general IS context [15]. Contrary to the two of the main relationships of the TAM, analysis of the findings suggests that perceived ease of use has insignificant effects on perceived usefulness while perceived usefulness could not mediate the influence of perceived ease use on intention which indicates the changing perceptions of users on the relevance of perceived ease of use on perceived usefulness. The contemporary users of IS now compared to IS users three decades ago, are likely to have better self-efficacy in technology due to the technological developments and increase in technology.

Based on the findings of this study, employees’ positive perceptions on the benefits of e-training and its ease of use are most likely to be of importance for successful implementation of e-training in the universities. This requires that priority should be accorded to the aspects of e-training that justifies the benefits of its usage. For instance, employees must be convinced that e-training can be easily used and offers them the opportunity to enhance their job performance and facilitate career development. Hence, management and supervisors should be involved in improving employees’ perceptions about using e-training in the universities. In addition, management of the universities should ensure that employees overcome the fear of potentially wasting time and disclosing sensitive information among employees. In this regard, the universities need to provide trust building supports, such as giving adequate pre-implementation training to employees, enhancing their computer and internet use skills, and ease at which they can use e-training system. Essentially, supported roles of supervisors can greatly enhance trust and employee support for e-training in the universities. Likewise, the influence of interactivity on e-training use intention suggests that employees will prefer interactive features of e-training system that enhances maximisation of the benefits of e-training and that which create positive disposition towards e-training use. Thus, employees are most likely to appreciate and use interactivity tools that they can have certain control on and that which provides prompt response to their requests and other queries. Managerially, importance should be accorded to e-training systems that provides most interactivity tools and functions. Adequate training should also be given to the users of e-training. Providers of e-training services should also focus on the aspects of e-training system that enhance interactivity. The design should emphasis user control, easy communication, and prompt responses to queries. Generally, trust and perceived usefulness should be given more priority prior to the implementation of e-training for having high performance and highest total effects on intention to use e-training respectively as indicated in the IPMA findings suggested.

### 4.3. Limitations 

While the contributions of this study to both theory and practice have been established, the study is not without some limitations that must be taken into consideration. First, this scholarly work was constrained by the fact that data collected and analysed came from employees of five federal universities of technology, where the majority of the respondents have at least a Master’s degree which could have their perceptions. A population that is well educated which might well have influenced the outcome of this study. Caution is thus warranted in generalising the findings of this study. Second, the study used an instrument adapted from the IS field and based on their application in private organisation whose validity and reliability were previously established. Second, considering the constraints of survey time, it was not possible for the researcher to include all possible factors influencing intention to use technology available in the extant literature. Third, although the instrument was further tested and validated in a pilot study and in the main survey, there is still the need for other factors associated with intention to be explored and potentially included in a more complete theoretical model. Fourth, the present study used a cross-sectional data for its analysis which inherently has limitations such providing just a snapshot perception, inability to measure some variables and uncover the meaning behind the data or causes and effects of variables.

### 4.4. Future Research

The limitations of this study provide a basis for conducting future research. Future research should consider conducting similar studies in a different environment and explore other factors influencing intention such as cultural lineage, group influence, and motivation in an e-training environment. Using qualitative and longitudinal studies could overcome the limitation of using cross-sectional data and to capture temporal aspects of e-training implementation.

## 5. Conclusions

This study deals with an important aspect of the IS literature through the prediction of employees’ intention to use e-training system by extending the TAM with interactivity and trust constructs. Previous findings have demonstrated the strong influences of interactivity and trust on intention [11,42]. Based on the analysis of the findings of this study, interactivity and trust are good predictors of intention in an e-training environment. The study has also established the relevance of the TAM’s belief factors of perceived ease of use and perceived usefulness in predicting intention to use e-training. The study has further demonstrated the applicability of TAM in predicting intention to use e-training in a pre-implementation and mandatory-use environment, and in the context of Nigeria.

## Figures and Tables

**Figure 1 behavsci-07-00047-f001:**
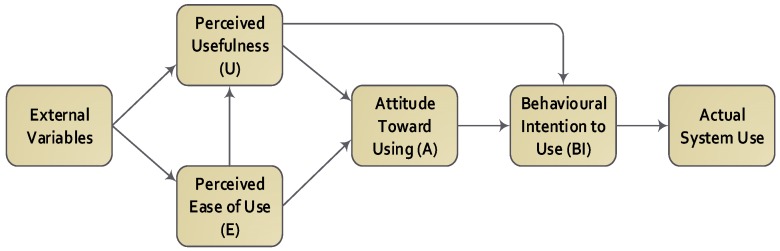
Technology Acceptance Model (TAM).

**Figure 2 behavsci-07-00047-f002:**
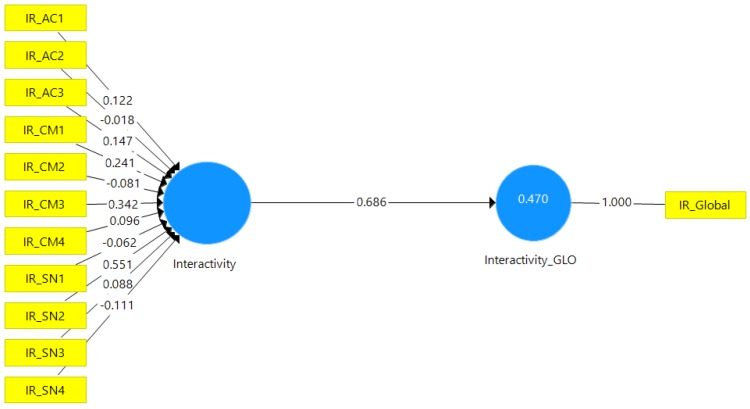
Formative Construct’s Convergent Validity.

**Figure 3 behavsci-07-00047-f003:**
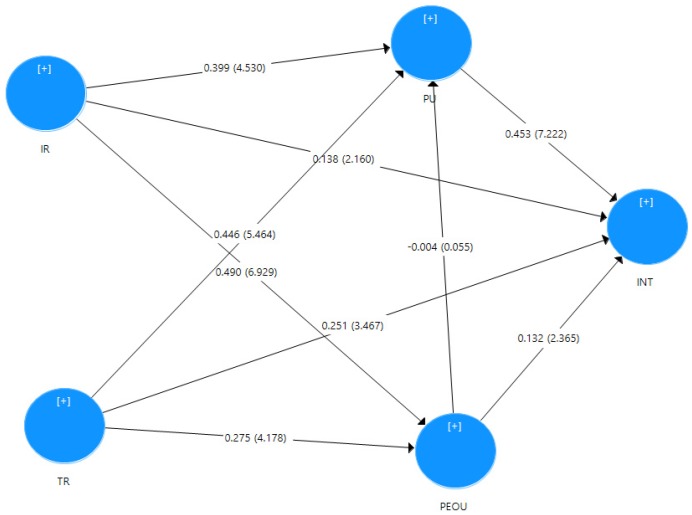
Structural Model.

**Table 1 behavsci-07-00047-t001:** Respondent Demographics.

Demographics	Features	Frequency	Percentage (%)
Gender	Male	117	54.9
	Female	97	45.1
	Total	214	100
Age	Below 29	42	19.6
	30–39	70	32.9
	40–49	66	30.9
	50 and above	36	16.6
	Total	214	100
Education	Secondary	9	4.3
	Diploma	21	10
	Bachelors’ Degree	41	18.9
	Master’s Degree	83	38.9
	Doctoral Degree	60	27.9
	Total	214	100
Staff Category	Academic	95	44.5
	Non-Academic	119	55.5
	Total	214	100.0
Computer/Internet Self-Efficacy Experience	<1 year	28	13
	1–3 years	75	35.2
	4–7 years	68	31.6
	8–11 years	29	13.6
	>12 years	14	6.6
	Total	214	100.0

**Table 2 behavsci-07-00047-t002:** Reflective Measurement Model Results.

Constructs	Items	Loadings	AVE	CR	Cronbach Alpha
INT	INT1	0.760	0.75	0.937	0.915
	INT2	0.849			
	INT3	0.898			
	INT4	0.914			
	INT5	0.899			
IRAC	IRAC1	0.825	0.704	0.877	0.792
	IRAC2	0.826			
	IRAC3	0.866			
IRCM	IRCM1	0.767	0.666	0.889	0.832
	IRCM2	0.817			
	IRCM3	0.870			
	IRCM4	0.808			
IRSN	IRSN1	0.769	0.608	0.861	0.785
	IRSN2	0.776			
	IRSN3	0.779			
	IRSN4	0.794			
PEOU	PEOU1	0.739	0.572	0.842	0.752
	PEOU2	0.780			
	PEOU3	0.741			
	PEOU4	0.763			
PU	PU1	0.700	0.677	0.926	0.903
	PU2	0.874			
	PU3	0.889			
	PU4	0.817			
	PU5	0.839			
	PU6	0.805			
TR	TRS1	0.769	0.622	0.908	0.879
	TRS2	0.806			
	TRS3	0.804			
	TRS4	0.806			
	TRS5	0.766			
	TRS6	0.782			

**Table 3 behavsci-07-00047-t003:** Discriminant Validity.

Fornell–Larcker Criterion								Heterotrait–Monotrait Ratio (HTMT)							
	INT	IRAC	IRCM	IRSN	PEOU	PU	TR		INT	IRAC	IRCM	IRSN	PEOU	PU	TR
INT	0.866							INT							
IRAC	0.578	0.839						IRAC	0.676						
IRCM	0.671	0.565	0.816					IRCM	0.772	0.675					
IRSN	0.485	0.514	0.574	0.780				IRSN	0.574	0.642	0.705				
PEOU	0.621	0.603	0.588	0.525	0.756			PEOU	0.742	0.758	0.744	0.665			
PU	0.800	0.481	0.693	0.502	0.537	0.823		PU	0.876	0.561	0.798	0.587	0.649		
TR	0.748	0.503	0.624	0.580	0.606	0.713	0.789	TR	0.833	0.591	0.723	0.699	0.732	0.793	

**Table 4 behavsci-07-00047-t004:** Formative Construct’s Properties.

Construct	Items	Convergent Validity	Weights	VIF	*t*-Value Weights	Sig
Interactivity	IRAC	0.70	0.341	1.594	4.887	0.000
	IRCM		0.640	1.748	9.711	0.000
	IRSN		0.180	1.619	2.479	0.013

**Table 5 behavsci-07-00047-t005:** Structural Model Results.

Relationships	Path Coefficient	S.E	*t* Values	*p* Values	5.0%	95.0%	Sig. Level	Decision
PEOU → INT	0.132	0.056	2.365	0.018	0.025	0.247	**	*Supported*
PEOU → PU	−0.004	0.068	0.055	0.956	−0.128	0.136	ns	*not supported*
PU → INT	0.453	0.063	7.222	0.000	0.325	0.571	***	*Supported*
IR → INT	0.138	0.064	2.160	0.031	0.010	0.259	**	*Supported*
IR → PU	0.399	0.088	4.530	0.000	0.234	0.577	***	*Supported*
IR → PEOU	0.490	0.071	6.929	0.000	0.351	0.631	***	*Supported*
TR → INT	0.251	0.072	3.467	0.001	0.107	0.387	***	*Supported*
TR → PU	0.446	0.082	5.464	0.000	0.269	0.590	***	*Supported*
TR → PEOU	0.275	0.066	4.178	0.000	0.145	0.406	***	*Supported*

* *p* < 0.01, ** *p* < 0.05, *** *p* < 0.01 level of significance; ns = not significant.

**Table 6 behavsci-07-00047-t006:** Mediation Results.

Relationships	Beta	S.E	*t* Values	*p* Values	2.50%	97.50%	Sig. Level	Decision
IR →PEOU → INT	0.062	0.032	1.938	0.053	0.010	0.134	*	Supported
IR → PU → INT	0.186	0.048	3.872	0.000	0.101	0.287	***	Supported
TR → PU → INT	0.198	0.045	4.364	0.000	0.107	0.285	***	Supported
TR → PEOU → INT	0.033	0.016	2.124	0.034	0.006	0.068	***	Supported
PEOU → PU → INT	−0.001	0.030	0.031	0.976	−0.058	0.058	ns	not supported

* *p* < 0.01, ** *p* < 0.05, *** *p* < 0.01 level of significance; ns = not significant.

**Table 7 behavsci-07-00047-t007:** Importance-Performance Map Analysis (IPMA).

Construct	Importance	Performance
IR	0.331	61.349
PEOU	0.152	73.770
PU	0.505	73.173
TR	0.588	76.617

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
