# Peer review of "Interactivity and Trust as Antecedents of E-Training Use Intention in Nigeria: A Structural Equation Modelling Approach"

_behavsci, 2017, doi:10.3390/bs7030047_

Round 1

Reviewer 1 Report

This paper reports a study carried out in five universities on employees' intention to use e-training based on the Technology Acceptance Model (TAM).

Suggestions for improvement include:

Interactivity needs an operational definition as it is not clear to the reader what this is, especially when given in the Abstract.

Not sure 'SmartPLS 3.0' needs to be mentioned in the Abstract.

Also, IPMA needs explanation here.

The focus is on e-training and the argument is given that there is little on this aspect of IT use in the literature, but there is a large number of publications on the TAM and many more could be included here: for example, the work of Timothy Teo.

Interactivity is described as a second order construct - not sure what this is, but a definition would have been good.

Small point, line 78, Liu's interactivity dimensions [19] is the correct order. Ditto, lines 85-86 - full stop after reference and no sentence.

There are 14 hypotheses which is rather a lot. I think I would have moved 'trust' closer to 'interactivity' in the order OR dealt with the TAM ones first, 'perceived ease of use', and 'perceived usefulness', and then the other two which are extensions of TAM.

Section 3: Materials and Methods

There is very little in this section on Materials, so why not call it Methods? And it would be good to see the questionnaire, perhaps in an Appendix.

3.1 Respondents - this was quite confusing - 364 respondents to 343 to 214? What's happening here?

Line 197 - do not need to state this about studying the whole population as this usually applies in experimental work.

Suggest present the demographics as tables (as this would help the reader).

In sum, I felt this section could be clearer.

Also, what about ethics? What procedures were followed? as these need to be mentioned.

Section 4. Results

It needs to be more evident that these are following the 14 hypotheses, which the reader will now not be able to remember!

p<0.000 should be avoided - always include a value more than zero.

Suggest provide more information about the IMPA, and its contribution to the analysis: for example, is there a reference for this other than Hair?

5.1 Discussion is generally fine - just needs a few minor tweaks. I would not refer to H1 but instead, give the hypothesis in full (as it makes it easier for the reader). There are four issues: perceived ease of use; perceived usefulness; interactivity; trust. The Discussion needs to mirror the Introduction (and the 14 hypotheses) and the Results in terms of the order of dealing with these aspects.

Line 356 - confirmed, not confirm

In 5.2, there needs to be a very clear delineation between what the results are showing and the implications of the findings. I would err on the side of caution here and not make claims - the paragraph beginning line 396.

Line 394 - needs checking re sense

In 5.3, I think I would separate out Limitations from Future Directions. The current paragraph is mainly dealing with the Limitations. A second one on Future Research would be good.

Line 421 - 'have' been established, not 'has'

Finally, where are the Conclusions?

Author Response

NO.

Reviewer's comments

Correction
Page
1
Operational definition for interactivityOperational definition of interactivity provided145 - 147
2
Including SmartPLS 3.0 not needed in the AbstractSmartPLS 3.0 has been removed15 - 16
3

IPMA needs explanation here

More explanation on IPMA has been provided

20 – 22

and

362 – 365
4
On publications on the TAM and the example of Timothy Teo’s workThe researchers argued that while a lot have been researched on TAM to investigate intention in the general area of IS, such use has been scanty in respect of e-training in public organisations especially within the context of the present study47 - 50
5
Meaning of interactivity as a second-order constructMore highlights on second-order construct (interactivity) have been provided

79 – 82

6
Line 78, Liu's interactivity dimensions [19] is the correct order. Ditto, lines 85-86 - full stop after reference and no sentenceThese have been corrected82 - 85
7
Move 'trust' closer to 'interactivity' in the order OR dealt with the TAM ones first, 'perceived ease of use', and 'perceived usefulnessThe TAM constructs were considered first and then followed by interactivity and trust constructs. These changes have also affected the hypotheses numbers.87 - 135
8
There is very little in this section on Materials, so why not call it Methods? And it would be good to see the questionnaire, perhaps in an Appendix.

The sub-title Materials and Methods has been changed to Methods.

Questionnaire attached as appendix A.

200

701 -711
9
Number of Respondents 364 to 343 to 214?The correct number of respondents (214) has been used204 - 206
10
Line 197 - do not need to state this about studying the whole populationStatement removed203
11
Present the demographics as tablesTable on Demographics provided220
12
What about ethics?Ethical considerations included223 - 229
13
It needs to be more evident that these are following the 14 hypothesesThe results have reorganised to follow the 14 hypotheses329 - 370
14
p<0.000 should be avoidedp < 0.000 avoided329 - 370
15
More information about the IMPA, and its contribution to the analysis: for example, is there a reference for this other than Hair?More information on IPMA provided382 - 386
16
The Discussion needs to mirror the Introduction (and the 14 hypotheses) and the Results in terms of the order of dealing with these aspectsThe discussion section has been reorganised to mirror introduction and the 14 hypotheses395 - 429
17
Line 356 - confirmed, not confirmCorrected to confirmed418
18
I would err on the side of caution here and not make claims - the paragraph beginning line 396Statement restated449 - 450
19
Line 394 - needs checking re senseCorrected by restating the statement 446 - 448
20
Separate out Limitations from Future Directions. The current paragraph is mainly dealing with the Limitations. A second one on Future Research would be goodFuture research separated from limitations and contents reorganised490 - 496
21
Line 421 - 'have' been established, not 'has'Corrected
475
22

Finally, where are the Conclusions?

Conclusions provided497 - 506

Reviewer 2 Report

This manuscript presents an analysis of the intention to use e-training among university employees in Nigeria using the TAM with interactivity and trust factors influencing employees’ intention to use e-training through perceived usefulness and perceived ease of use. The authors report that the critical role of interactivity and trust as significant determinants of intention to use e-training. The results of the empirical analysis show that interactivity has significant influence on intention, on perceived ease of use and on perceived usefulness. The results of the Importance-Performance Map Analysis show, that trust is most important construct influencing intention to use e-training. The authors also interpret their results. The conclusions are mostly well supported by the results.

The manuscript is written on a good body of research on this topic. The manuscript is well written and the theoretical foundation using the TAM-Model and it underpinning theory is also relevant and interesting. The study is relevant because the central question about the roles of interactivity and trust in e-training intention is of general relevance. The study has a good design and the description of the methods and results is clear and easy to follow.

I have some comments.

The study analyzes the intention to use e-training among well-educated university employees in Nigeria. How is the impact of the Nigerian individual culture dimensions (Hofstede) and the educational background of the respondents on the research results?

Other studies show that that individual characteristics and technological factors may have a significant influence on intention to use informational electronic systems (Ronnie H. Shroff, Christopher C. Deneen and Eugenia M. W.  Analysis of the technology acceptance model in examining students’ behavioural intention to use an e-portfolio system, Australasian Journal of Educational Technology 2011, 27(4), 600-618).

Additional comments for authors

There are various typos and minor errors, which I have not listed here (e.g. in line 296).

A part of sentence in Line 86 is missing.

I could find no mention of WHEN the data were collected.

Author Response

SN
Reviewer's comments
Correction
Page
1
How is the impact of the Nigerian individual culture dimensions (Hofstede) and the educational background of the respondents on the research results?

The present study considered only interactivity and trust as external determinants of intention to e-training because the literature suggest that the two constructs were mainly applied in study relating to e-commerce, e-banking, etc thereby overlooking their relevance in e-training.

The impact of the educational background of the respondents has been incorporated as recommended

364 – 365

425 - 427

2
Other studies show that that individual characteristics and technological factors may have a significant influence on intention to use informational electronic systems (Ronnie H. Shroff, Christopher C. Deneen and Eugenia M. W.  Analysis of the technology acceptance model in examining students’ behavioural intention to use an e-portfolio system, Australasian Journal of Educational Technology 2011, 27(4), 600-618).The main focus of the present study is on the roles of interactivity and trust on intention to use e-training which stems from the fact that literature suggests that the influences of these constructs on intention to use e-training were yet to be established although they have been applied in the general IS literature41 - 50
3
There are various typos and minor errors, which I have not listed here (e.g. in line 296).Typos identified and corrected288, 296, 336, 355, and 421
4
A part of sentence in Line 86 is missingThe sentence has been corrected85 - 86
5
I could find no mention of WHEN the data were collectedPeriod of data collection included222 - 223
